# An Exploration of People Living with Parkinson’s Experience of Cardio-Drumming; Parkinson’s Beats: A Qualitative Phenomenological Study

**DOI:** 10.3390/ijerph21040514

**Published:** 2024-04-22

**Authors:** J. Yoon Irons, Alison Williams, Jo Holland, Julie Jones

**Affiliations:** 1School of Psychology, College of Health, Psychology and Social Care, University of Derby, Derby DE22 1GB, UK; 2Parkinson’s Scotland Office, 1/14 King James VI Business Centre, Friarton Road, Perth PH2 8DY, UK; alison@edinburghparkinsons.org (A.W.); jhollandparkinsons@gmail.com (J.H.); 3School of Health Sciences, Robert Gordon University, Garthdee Road, Aberdeen AB10 7QG, UK; j.c.jones@rgu.ac.uk

**Keywords:** Parkinson’s disease, drumming exercises, flow experience, empowerment, physical activity, mental health, eye-hand coordination, multi-disciplinary, music, healthy ageing

## Abstract

Research has shown that physical activity has a range of benefits for people living with Parkinson’s (PLwP), improving muscle strength, balance, flexibility, and walking, as well as non-motor symptoms such as mood. Parkinson’s Beats is a form of cardio-drumming, specifically adapted for PLwP, and requires no previous experience nor skills. Nineteen PLwP (aged between 55 and 80) took part in the regular Parkinson’s Beats sessions in-person or online. Focus group discussions took place after twelve weeks to understand the impacts of Parkinson’s Beats. Through the framework analysis, six themes and fifteen subthemes were generated. Participants reported a range of benefits of cardio-drumming, including improved fitness and movement, positive mood, the flow experience, and enhanced social wellbeing. A few barriers to participation were also reported. Future research is justified, and best practice guidelines are needed to inform healthcare professionals, PLwP and their care givers.

## 1. Introduction

Parkinson’s is the fastest-growing neurodegenerative condition [1]. Ten million people live with Parkinson’s worldwide [2], and over 145,000 people in the UK are diagnosed with Parkinson’s [3]. Due to the ageing population, the incidence of Parkinson’s is expected to continue to rise [4], highlighting the need for effective healthcare interventions to support a growing global community. 

There is no cure for Parkinson’s, and its exact cause remains unknown. However, increased age, lifestyle, and a complex interaction of environmental and genetic factors are commonly implicated [5]. Parkinson’s is characterised by the loss of dopaminergic neurones within the Substantia Nigra Pars Compacta. The progressive loss of neurones results in a decrease in activity of the Nigrostriatal circuits, which increasingly inhibits the Basal Ganglia. Ultimately, this leads to a decrease in or poverty of movement, which is synonymous with Parkinson’s. The exact mechanism precipitating neuronal loss in Parkinson’s is unknown. Due to the progressive loss of neurones, Parkinson’s is now recognised as a broad multi-system condition encompassing over 40 motor and non-motor symptoms [6]. Historically, Parkinson’s was regarded solely as a movement disorder [7]; however, non-motor symptoms are now widely recognised as integral to Parkinson’s, with apathy, depression, and anxiety being prevalent among the Parkinson’s community [4]. 

The management of Parkinson’s is complex, owing to its progressive nature, patient heterogeneity, and symptom diversity. Management is reliant upon medication, which targets dopamine imbalance through a variety of different mechanisms [8]. Moreover, medication efficacy is time limited, with Parkinson’s progression necessitating different combinations of medications, taken at increasing dosages, due to the medication’s effect wearing off. Thus, with a growing Parkinson’s population, and finite benefits from medication, there is a pressing need to develop effective long-term health interventions.

Physical activity (PA) has been hailed as “the new medicine” for Parkinson’s; PA is no longer viewed as a complementary intervention, but of equal importance to medication [9]. The interest in PA has been fuelled by the association between PA and the reduced risk of developing Parkinson’s [10] and the potential to attenuate symptom progression [11,12]. Systematic reviews highlight that PA results in improved strength, balance, gait, and physical capacity [13,14,15,16], as well as improved motor and non-motor symptoms [17,18,19,20,21,22]. Current guidelines advocate that weekly PA programmes should be prescribed in a progressive manner, including strength, balance, aerobic, gait, and task-specific training focusing on the upper and lower limbs and spine, with emphasis placed on functional movement patterns and large amplitude movements [23]. In addition, the high incidence of apathy in Parkinson’s means that to maintain motivation, PA needs to be enjoyable.

### Parkinson’s Beats

Cardio-drumming programmes or drumming exercises have recently been gaining popularity. Drumming is a relatively high-intensity aerobic exercise involving whole-body movements [24]. A workout ball in a bucket (e.g., [25]), or African drums (e.g., [26]) can be used, as they are easily available and do not require previous training or specific knowledge and skills to play. Drumming exercises combining movement with rhythmic, up-beat music, build enjoyment and support the incremental pace, aiming to increase cardiovascular benefits. 

The core of drumming relies on our intrinsic ability, known as entrainment, to tap/play along with the rhythms of music (i.e., synchronising our movement with music/beats). This entrainment is of particular importance for people living with Parkinson’s (PLwP), as the music’s beats can effectively support the brain to coordinate movements [27,28]. For example, a systematic review of meta-analyses of clinical studies involving music for movement rehabilitation in Parkinson’s demonstrated that when music was added to gait training, it resulted in significant improvements of balance, stride length, and walking (measured using the Timed Up and Go test) in PLwP [29]. This is because rhythmic stimulation offers beneficial time-based cues for the brain to plan and be ready to execute the next movement [27]. Applying rhythmic entrainment to drumming exercises, which involve synchronised movements stimulated by beats, Parkinson’s Beats is specifically adapted for PLwP to promote large amplitude arm movements, balance through differing drumming techniques, and strength training in the legs. Further, a randomised controlled study highlighted that group-based drumming programmes decreased participants’ depression and anxiety, and increased wellbeing and social resilience [30]. Moreover, positive experiences including enjoyment, a sense of control, accomplishments, and social connectedness were reported in a study with mental health service users [31]. To utilise these numerous advantages of drumming, a pilot study was conducted involving eighteen PLwP, where only eight participants took part in drumming and ten participants were in the control group [32]. Following the twice-weekly 6-week West African drum circle intervention, participants in the drumming group demonstrated a statistically significant improvement in the quality-of-life measure (PDQ-39), but no improvements in depression and motor function assessments. In this study, the qualitative assessment was limited to several quotes from participants’ diary entries, and no formal qualitative analysis was involved. Therefore, the current study aimed to explore the impacts of a cardio-drumming programme, Parkinson’s Beats, which uses a workout ball in a bucket with two sticks. Given the fact that this was only a recently adopted exercise programme for PLwP, the focus of this study was to gain a better understanding of the lived experience of PLwP taking part in Parkinson’s Beats using a descriptive phenomenological method [33]. 

## 2. Methods

### 2.1. Design

A qualitative phenomenological research design [34] was used to gain an understanding of the impact of Parkinson’s Beats (PB). Adopting a phenomenological approach allows for a greater understanding of and insight into the experience of PLwP who participated in Parkinson’s Beats. The descriptive phenomenological approach seeks to explain the nature of things through the way people experience them [34]. Such an approach allows us to capture and describe essential aspects of the participants’ experience, to understand their experience in their context, i.e., living with Parkinson’s [33]. This study was ethically approved by the Robert Gordon University School of Health Sciences Research Ethics Committee (SREC SHS/22/32). 

### 2.2. Participants

This study adopted a convenience approach to sampling, inviting PLwP who had attended four or more Parkinson’s Beats sessions, delivered either online or face-to-face, to participate in a focus group to explore their experiences of participation in Parkinson’s Beats. As Parkinson’s Beats is an inclusive form of activity and can be adapted to suit all abilities, no restriction was based on stage of Parkinson’s, time since diagnosis, or Parkinson’s medication. Parkinson’s Beats attendees received a participation information sheet via email and provided statements of informed consent prior to participating in the study. Although we did not formally assess eligible participants on their cognitive function, all our participants did not have any form of dementia. They provided statements of informed consent, followed all instructions during the weekly cardio-drumming sessions, and were able to share their thoughts and experiences of Parkinson’s Beats at the focus group discussions.

### 2.3. Data Collection

The qualitative data were gathered through audio-recorded conversations of three focus groups: one in-person group (*n* = 9) and two online groups via Microsoft Teams (*n* = 5 in each group). Focus groups 1 and 2 were conducted immediately following the drumming session; the third was held on a day where there was no drumming.

Focus group discussions were guided by a standardised topic guide incorporating four key areas of interest: (i) their initial thoughts of Parkinson’s Beats, (ii) their experience of Parkinson’s Beats, (iii) the perceived impact of Parkinson’s Beats, and (iv) barriers and motivators to participation in Parkinson’s Beats.

All focus groups were facilitated by the same researcher (AW). An additional member of the research team (JH) served as an observer and took field notes to complement the qualitative analysis. All focus groups were recorded, and accurately transcribed. 

### 2.4. Data Analysis

Interview transcripts were analysed following Ritchie and Lewis’ framework analysis approach [35], which is commonly used within health-related research [36], and is ideally suited to exploring participants’ perceptions, experiences, and values, aligning with the objectives of this research. The framework analysis approach was selected as it offers a flexible approach to identify and describe the data and categorise key patterns to generate themes related to the phenomenon of interest [34]. Using this inductive approach meant that participants’ views and opinions dictated the emergence of themes and subthemes. The researchers (AW and JH) grouped themes and subthemes, discussing their scope and nature with the rest of the research team (JJ and JYI), and making any necessary amendments. 

## 3. Results

In total, seven men and twelve women, ranging in age between 55 and 80 years old, took part in the Parkinson’s Beats study. Participants (*n* = 5) were excluded from the analysis only if they had attended fewer than four sessions of Parkinson’s Beats.

The framework analysis resulted in the identification of six themes and fifteen subthemes. Themes centred upon physical improvements, benefits on mood, cognitive function, social wellbeing, additional benefits, and barriers to participation, as illustrated in Table 1.

### 3.1. Theme 1. Physical Impact

The focus groups were intentionally run following a Parkinson’s Beats session, to capture participants’ immediate responses. Experiences of Parkinson’s Beats varied from some participants feeling euphoric and energised; further, some participants perceived that they had a good work out.

“*I am just buzzing after attending the class, like I am ready to take on anything*”

“*I feel energised, and it lasts the rest of the day*”

“*Done some exercise, love the music, love instructor. It’s a happy class. You feel you’ve achieved something as well*”

Some participants reported “muscle tiredness” and “some muscle ache” but this was not expressed negatively; rather, these comments attested to the fact that they had a good workout. However, one participant reported that they could no longer do both Parkinson’s Beats and Tango class in the same day again, suggesting that Parkinson’s Beats was a high-intensity exercise.

In addition to physical benefits, participants also reported functional benefits. These included, but were not limited to, an improved ability to raise their arms with greater ease, so that they were now able to reach top shelves. Improvements to upper limbs were also evidenced by participants reporting a greater ability to use their arms to get out of bed more easily, and increased mobility. In addition, participants also reported perceived reductions in pain, stiffness, and tremors on drumming class days.

“*Yes, I’m like that—I can reach cupboard shelves that I couldn’t reach before*”

“*When I first started doing this—I have a lot of pain with my PD [Parkinson’s disease], I have arthritis and I have a lot of issues. I struggle to get up and down the stairs. So I used to get the lift up to my class and the lift down. So about the fourth session I came out of the class and I was chatting away to people and I was at the bottom of the stairs before I realised I had walked down the stairs and I hadn’t felt any pain. And it’s just the exhilaration of the drumming that had done that*”

The benefits of drumming also appeared to have a lasting effect with a positive impact continuing for the rest of the day and a carry-over effect into other activities. Several participants reported improved hand/eye coordination, improved handwriting, and greater amplitude of movement, which they attributed to Parkinson’s Beats.

“*I am not sure if it is because of the drumming, but I have notice my hand writing has improved recently, and my medication has not changed*”

Even though the physical benefits seemed small, they made a big difference. Many reported that they were able to do tasks which they could not do before. Within the focus groups, it became apparent that when they started talking and sharing experiences about how Parkinson’s Beats had impacted them, they realised what the changes had been. Enhanced physical and functional benefits were also perceived to fuel changes in mood.

“*On drumming days I’m a lot more [chilled] because I don’t have the pain and the stiffness*”

“*I would say, if you want to feel good, and not to feel like you do first thing in the morning, when you need to stretch and it takes half an hour or an hour to get yourself sorted, come to a drumming class. The simple action of using your arms and your hand/eye coordination. It really lifts you body and mind*”

### 3.2. Theme 2. Emotional Impact

Parkinson’s Beats was unanimously perceived to have a positive impact on these non-motor symptoms: In terms of immediate impacts, the sessions, participants said, are “Happy” and “It’s a happy class”. They used the word “euphoria” and noticed that their mood on ‘drumming days’ was much better than on others: 

“*Upbeat—you do feel good. You feel up rather than down*”.

“*Parkinson’s Beats exercise drumming makes me happy*!”

Attending Parkinson’s Beats also served as a distraction for many, allowing them to forget about their Parkinson’s disease for a period of time:

“*Actually doing it, I forget about the Parkinson’s”; and “I forget about Parkinson’s for the rest of that day*”.

“*It’s amazingly relaxing, the whole thing; and you’re concentrating on something other than your tremor. You’re concentrating on the drumming and forget about your tremor*”

Such a distraction appeared to help to mask the daily pain and frustration normally experienced by PLwP. The focus groups all reported that PB drumming helped to reduce stress and improve emotional wellbeing. The impact of music, the act of drumming, and the fun environment created by the instructor were reported as factors which aided the positive impact on the emotional well-being of the group:

“*I just love it—it’s a selfish thing for me, I love to see people enjoying themselves—it gives so much back to me*”

“*This is fun—if you can find something you can enjoy then you’re more likely to keep it up. She [the instructor] makes it fun, she really does*”

Further, a sense of empowerment is also remarked on by participants. Parkinson’s Beats promoted “a sense of control”, and inclusivity.

“*Just enjoy what you can do. Every day is different. Some days you can’t do a lot*”

“*You don’t have to—that’s the beauty of drumming, you are a free spirit, you do what you like. The more you do it, the better you get as well*”.

### 3.3. Theme 3. Impact on Cognitive Function

Participating in Parkinson’s Beats presented a cognitive challenge for participants. Participants reported that they enjoyed the challenge of the complex tracks and rhythms:

“*I like the more complex [music] tracks” and “There’s more for you to do, it’s more entertaining, you feel more positive at the end of it*”

“*You’re pushing yourself more as well because some of the more complex movements at more difficult to achieve*”

“*You have to concentrate so much, and quite a lot of effort put into it*”.

Participants also found great satisfaction in stretching their skills.

“*Pleased with myself, I’ve accomplished the session” and “Maybe smug would be a better word than content*”.

However, some participants reported frustrations around balancing between their skills and the challenge of complex tracks and movements:

“*That’s really important that people don’t feel that they have to achieve a standard. Cos there is no standard*”

“*It just makes me so cross that I can’t do it as well as I would like to*”

### 3.4. Theme 4. Social Impact

The participants had a positive experience of being in a group, enjoyment of making music together, and working with others in a group. Parkinson’s Beats appeared to have brough a group of PLwP with their limitations all together.

“*Trying to finish at the same time as everyone else” and “Being in tune with others is a great feeling*”

“*The feeling when you’re all banging at the same time is great*”.

“*Participants were clear that “Doing it [drumming] on your own wouldn’t be the same*”.

### 3.5. Theme 5. Additional Benefits

Three subthemes emerged in this theme: music, dance, and the loss of self-consciousness.

Music appears to be the essential impetus for the drumming exercises. Participants felt strongly that they were “Joining in, making [their] own contribution to the track”, in effect playing music with the musicians on the recording. “There’s something about creating music, dancing to the beat, or drumming to the beat”. They are referencing entrainment. The awareness of music permeates the rest of the day, and beyond. “It’s funny—I do listen to music in a different way now. I’m thinking: How would I move/beat to that music? you know, when you’re in the car, and you bang the steering wheel …”

Additionally, some participants incorporated drumming with dancing and movement. The unique blend of uplifting music and aerobic-alike exercises appeared to support dancing and body movements.

“*It’s more like dancing for me—there’s movement in it, and I tend to try to make it into a dance anyway*”

Moreover, there were some unexpected benefits for online participants: “I like the zoom. I’m not so sure I’d do it if it wasn’t on zoom. I’d be so self-conscious, that I was not hitting the drum at the same time that it would put me off” Offering the Parkinson’s Beats sessions online also enabled people to gain benefits: “I like to be able to ‘go for it”; “I like being able to lose all of my frustrations”; “Just start making your own noise. Fun. Lash out”; “You can hit the ball as hard as you like”.

### 3.6. Theme 6. Barriers to Participation in Parkinson’s Beats

Three potential barriers emerged in the focus group discussions: self-consciousness, retrospection, and the music chosen.

Self-consciousness was overcome through the online setting. One of the online participants said: “I like the zoom. I’m not so sure I’d do it if it wasn’t on zoom. I’d be so self-conscious, that I was not hitting the drum at the same time [as the others] that it would put me off”.

Another barrier that emerged was raised by a newly diagnosed participant who was frustrated with their own limited mobility and subsequent lack of precision and acknowledged that “I can’t forget about my illness when I’m drumming, because it just make me so cross that I can’t do it as well as I would like to. [Told, there’s no pressure, the participant responded] the pressure is all at my end”.

Moreover, a barrier arose for online participants when drumming at home—the impact of the noise on other members of the household. “I have a problem—we now have a rabbit living in the house so now I have to quieten down a bit—the rabbit doesn’t like it” and “It can be a problem, the noise. My husband would agree”.

Given that part of people’s enjoyment of Parkinson’s Beats is “I like to be able to ‘go for it” and “…lose all of my frustrations” and “making your own noise. Lash out” by hitting the ball “as hard as you like”, these restrictions have to be taken into account by the facilitator and the online participants affected.

And finally, the music chosen for the classes was off-putting for some: “A couple of people came but didn’t like the music. Said if it was classical, they would be more likely to join”.

## 4. Discussion

This study aimed to explore the perceived impact of cardio-drumming adapted specifically for PLwP using a descriptive phenomenological approach. Six themes and fifteen sub-themes were generated, which highlighted the physical, emotional, cognitive, and social benefits as well as barriers to participating in Parkinson’s Beats. It is of interest that some of the perceived benefits were both immediate and lasting over the course of programmes.

Current guidelines recommend a multimodal approach to exercise, encompassing strength, flexibility, balance, and aerobic exercise [37]. The physical benefits reported by participants in Parkinson’s Beats may be attributed the fact that drumming combines all these elements within a single workout. Participants also reported perceived improvements in balance following participation in Parkinson’s Beats. Improved balance could be attributed to the dual tasking component of Parkinson’s Beats whereby drumming rhythmically to the music demands both physical and cognitive resources. Dual tasking involves undertaking two tasks simultaneously: for example, a combination of two motor tasks or a combination of a motor and cognitive task [38]. Several authors have reported that among PLwP the secondary task, regardless of whether it is motor or cognitive, impairs the performance of the primary task [39]. Being able to do two tasks simultaneously is important during many functional tasks such as walking and talking. Consequently, dual task training is advocated in the European Parkinson’s Physiotherapy guidelines [40]. A recent systematic review conducted by Beline De Freitas and colleagues [41] demonstrated that dual task training resulted in gait and balance benefits. This study was not designed to explore the effectiveness of Parkinson’s Beats on gait and balance, but these perceived benefits may warrant further exploration in a future study.

Additionally, Parkinson’s Beats participants reported a range of emotional benefits including enjoyment and the experience of being in a flow. Participants enjoyed having uplifting well-known songs, having the group support, and feeling achievement, as well as being able to ‘forget’ their Parkinson’s during the sessions. These findings were similar to those of previous drumming studies: an in-depth qualitative study involving mental health service users highlighted that the drumming programme appeared to have engendered mental health recovery in the participants [31], as the participants demonstrated enhanced enjoyment, agency, accomplishments, engagement, and positive self-identity [31]. In particular, the Parkinson’s Beats study participants also reported experiences of being in a flow, which is characterised by being totally immersed in the joy of the activity; this experience of being in a flow has been conceptualised by Csikszentmihalyi [42,43]. Being in a flow can be highly motivational, as we experience heightened enjoyment of an activity that balances challenge and stretches our ability to meet and surpass those challenges [42,43]. This is evident in our study participants, PLwP, who were motivated by their sense of happiness, exhilaration, and outright euphoria. Similar findings were reported in an African drumming study involving older adults (N = 27) with dementia, who showed improvements in mood, level of interest, responsiveness, engagement, and enjoyment [26]. In another controlled study, a three-month Japanese drumming exercise programme resulted in reduced depressive mood and improved physical fitness in community-dwelling older women (≥65 years) (N = 40) [44]. Further, a randomised controlled study demonstrated significantly reduced depression and anxiety and increased wellbeing and social resilience in the drumming group participants compared with the control group [30]. This study also tested participants’ saliva samples and found that drumming resulted in increases in anti-inflammatory activity [30].

Moreover, a sense of empowerment was also remarked on by Parkinson’s Beats participants. Empowerment is variously described as “a process whereby people gain mastery over their affairs” [45]; “helping patients discover and develop inherent capacity to be responsible for one’s own life” [46]; and a sense of power and control of one’s condition. Rawlet [47] proposes that empowerment is determined by one’s self-efficacy. Levels of empowerment are associated with levels of quality of life: when PLwP have the capacity to control their own management and take personal responsibility for their own well-being, this can be empowering and yields an enhanced quality of life despite the seemingly limiting condition [48].

Due to Parkinson’s, coordinating one’s movements becomes increasingly difficult; however, Parkinson’s Beats participants reported improved skills and concentration (Theme 3). With the help of strong beats in the accompanying songs, the participants perceived there were benefits for the brain. Such cognitive benefits were evident in a 15-week controlled drumming study with older adults (N = 24), who felt their cognitive functions were declining [49]. This study was designed to progress from simple tasks to more complex tasks including drumming and singing for older adults. After the 15-week programme, the drumming group participants demonstrated improved visual memory compared with the literary control group. Thus, this study indicated that drumming programmes can potentially prevent cognitive declining [49].

Moreover, Parkinson’s Beats is inclusive because the exercises can be adapted to accommodate individual limitations and capacities. For example, PLwP can use one arm or two, be standing or seated; most will drum in real time, but the beat can be slowed to half time for anyone finding difficulty with the speed and frequency of the rhythm. Each session was led by an experienced facilitator who created an atmosphere of trust and safety, enabling participants to become free to express themselves, and enjoy themselves. The facilitator demonstrated enthusiasm and provided encouragement throughout the sessions. Additionally, our study demonstrated that Parkinson’s Beats can be delivered both face to face and online, where participants’ experiences were similar, although in some cases, the online format was preferred due to the option of not being seen/heard, which helped overcome self-consciousness. However, online participation, i.e., drumming at home may require some pre-arrangements with family members, who may not enjoy the noise. Further, the facilitators are required to have sound knowledge and understanding of Parkinson’s and its progressive impacts on participants. As we highlighted in Theme 6, the facilitator may need to provide extra support for those who are newly diagnosed, as the newly diagnosed participants may experience much frustration when they face their own limited mobility and subsequent lack of precision during drumming. Finally, collaborating on the song choices with participants could minimise disappointment or disagreement, as well as encourage participants to take ownership.

### Limitations and Strengths

This study was an exploratory pilot research study, which highlighted the experiences of PLwP who participated in cardio-drumming exercises. This study is based on responses of 19 participants who were interviewed in three focus groups. Arguably, a larger sample size could potentially yield richer data; however, the sample size in the current study aligns with prior Parkinson’s qualitative research. Guest, Bunce, and Johnson [50] found that across a sample size of sixty interviews, category saturation occurred within the first twelve interviews and that the basic elements for core themes were present within the first six interviews. Therefore, the data gathered from 19 participants in the current study were regarded as sufficient for the purpose of this study.

Further, we acknowledge that the researchers conducting the focus groups (AW and JH) have experience of drumming and they themselves live with Parkinson’s. While this may have the potential to introduce bias, the fact that they have lived experience of Parkinson’s may have allowed the collection of richer data, as participants may have felt more at ease during the focus groups. While the use of researchers who have Parkinson’s may have introduced some bias, this was mitigated during the data analysis, which was conducted by all members of the research team.

## 5. Conclusions

The findings of Parkinson’s Beats, a cardio-drumming exercise programme for people living with Parkinson’s (PLwP), has demonstrated physical, emotional, cognitive, and social benefits. PLwP reported enhanced mobility and reach, as well as reduced stiffness and pain. A range of positive feelings were evident, including feeling euphoric, uplifted, and joyful. Additionally, participants experienced flow and empowerment. Group sessions also enhanced social wellbeing. Future studies may include physical, cognitive, and psychological assessments, and investigate both short-term and long-term impacts, with appropriate comparators (e.g., other types of physical activities, online vs in-person delivery, healthy controls). Research into the optimal dose of cardio-drumming for different stages of Parkinson’s progression is also needed. Moreover, best practice guidelines for the facilitators/instructors on how to deliver a cardio-drumming exercise programme tailored for PLwP need to be developed using evidence-based research. Further research should also explore the possibility of Parkinson’s Beats being prescribed as an integral part of the personalised health plans of PLwP.

## Figures and Tables

**Table 1 ijerph-21-00514-t001:** Themes and subthemes of participants experiences.

Themes	Subthemes with Representative Quotes
Physical impact The focus groups were intentionally run following a PB session, to capture participants’ immediate respones.	1.1 Immediate post-class physical impact “*I am just buzzing after attending the class, like I am ready to take on anything*” “*I feel energised, and it lasts the rest of the day*” 1.2 Functional benefits “*Yes, I’m like that—I can reach cupboard shelves that I couldn’t reach before*” “*When I first started doing this—I have a lot of pain with my PD [Parkinson’s disease], I have arthritis and I have a lot of issues. I struggle to get up and down the stairs. So I used to get the lift up to my class and the lift down. So about the fourth session I came out of the class and I was chatting away to people and I was at the bottom of the stairs before I realised I had walked down the stairs and I hadn’t felt any pain. And it’s just the exhilaration of the drumming that had done that*” 1.3 Lasting effects “*I am not sure if it is because of the drumming, but I have notice my hand writing has improved recently, and my medication has not changed*” “*On drumming days I’m a lot more [chilled] because I don’t have the pain and the stiffness*” “*I would say the simple action of using your arms and your hand/eye coordination. It really lifts you body and mind*”
2. Emotional impact Parkinson’s Beats was unanimously perceived to have a positive impact on non-motor symptoms.	2.1 Impact during and immediately after class “*Upbeat—you do feel good. You feel up rather than down*”. Attending Parkinson’s Beats also served as a distraction for many, allowing them to forget about their Parkinson’s for a period of time: “*Actually doing it, I forget about the Parkinson’s”; and “I forget about Parkinson’s for the rest of that day*”. “*It’s amazingly relaxing, the whole thing; and you’re concentrating on something other than your tremor. You’re concentrating on the drumming and forget about your tremor*” 2.2 Distraction and reduced stress “*I just love it—it’s a selfish thing for me, I love to see people enjoying themselves—it gives so much back to me*” “*This is fun—if you can find something you can enjoy then you’re more likely to keep it up. She [the instructor] makes it fun, she really does*” 2.3 Enhanced emotional wellbeing and empowerment “*Just enjoy what you can do. Every day is different. Some days you can’t do a lot*” “*You don’t have to—that’s the beauty of drumming, you are a free spirit, you do what you like. The more you do it, the better you get as well*”.
3. Impact on cognitive function Participating in Parkinson’s Beats presented a cognitive challenge for participants. Participants reported that they enjoyed the challenge of the complex tracks and rhythms.	3.1 Mastering challenges “*I like the more complex [music] tracks” “There’s more for you to do, it’s more enter-taining, you feel more positive at the end of it*” “*You’re pushing yourself more as well because some of the more complex movements at more difficult to achieve*” “*You have to concentrate so much, and quite a lot of effort put into it*”. 3.2 Frustrations “*That’s really important that people don’t feel that they have to achieve a standard. Cos there is no standard*” “*It just makes me so cross that I can’t do it as well as I would like to*”
4. Social Benefits The participants had positive experiences of being in a group, enjoyment of making music together, and working with others in a group.	4.1 Working with others “*Trying to finish at the same time as everyone else*” “*Being in tune with others is a great feeling*” Participants were clear that “*Doing it [drumming] on your own wouldn’t be the same*”
5. Additional benefits Three subthemes emerged in this theme: music, dance, and the loss of self-consciousness.	5.1 Music “*There’s something about creating music, dancing to the beat, or drumming to the beat*” “*It’s funny—I do listen to music in a different way now. I’m thinking: How would I move/beat to that music? you know, when you’re in the car, and you bang the steering wheel …*” 5.2 Dance “*It’s more like dancing for me—there’s movement in it, and I tend to try to make it into a dance anyway*” 5.3 Losing self-consciousness “*I like the zoom. I’m not so sure I’d do it if it wasn’t on zoom. I’d be so self-conscious, that I was not hitting the drum at the same time that it would put me off*” “*I like being able to lose all of my frustrations*” “*Just start making your own noise. Fun. Lash out*”
6. Barriers Three potential barriers emerged in the focus group discussions: self-consciousness, retrospection, and the music chosen.	6.1 Self-consciousness “*I can’t forget about my illness when I’m drumming, because it just make me so cross that I can’t do it as well as I would like to. [Told, there’s no pressure, the participant responded] the pressure is all at my end*”. 6.2 Drumming noise “*I have a problem—we now have a rabbit living in the house so now I have to quieten down a bit—the rabbit doesn’t like it*” 6.3 Music choice “*A couple of people came once but didn’t like the music. Said if it was classical, they would be more likely to join*”

## Data Availability

The data are not publicly available due to ethical restrictions. The data can be obtained from the corresponding author based on a reasonable request for research purposes.

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
