# Peer review of "An Exploration of People Living with Parkinson’s Experience of Cardio-Drumming; Parkinson’s Beats: A Qualitative Phenomenological Study"

_ijerph, 2024, doi:10.3390/ijerph21040514_

Round 1

Reviewer 1 Report (New Reviewer)

Comments and Suggestions for Authors

Dear authors,

Thank you for the opportunity to review your paper and I enjoyed reading your article. You have clearly identified a gap in current body of knowledge regarding cardio drumming and Parkinson's disease. 

I have a few suggestions which could strengthen the paper.

1. Line 43 You could remove no cure exists for Parkinson's, as this has been mentioned before.

2. Using Richie and Lewis framework Analysis Approach appears not to be in line with a phenomenological study design. Within this approach Hycner's (1999) explicitation process is often used.  Your research appears s to be pragmatic in nature. An explanation of how a phenomenological study links with Richie and Lewis framework Analysis Approach would provide more clarity.

Line 157 In additional should be in addition. 

Line 269 - 270   should be in italics.

Line 290 qualitative approaches should be singular.

Some of the text is in bold and in red. I would like to suggest removing the bolded and red text colour and have it in line with the other text in the paper.

The study from a qualitative point of view is well carried out and the analysis of the data make sense. Because the data is derived from a couple of focus groups, you could argue that more data is required to underpin the findings. The notion of reaching data sufficiency is at play here. A larger mixed method study would probably suit your journal better. You also can’t rule out bias in this study as two of the researchers were also participants, as this might have skewed the findings.

Comments on the Quality of English Language

This is a well written paper. 

Author Response

Many thanks for your comments. We made amendments according to your suggestions. We hope our manuscript is now much improved. Thank you very much.

Reviewer 2 Report (New Reviewer)

Comments and Suggestions for Authors

Dear authors, your document entitled “An exploration of people living with Parkinson’s experience of cardio-drumming, Parkinson’s Beats.” is original and very interesting.

The following suggestions aims to improve the overall quality of the document.

First I would suggest that you include the type of analysis “A qualitative phenomenological research” in the title.

In the abstract section, you have mentioned 19 participants, while in the data collection, we find the following information “The qualitative data were gathered through audio recorded conversations of three focus groups: one group (n=5) in person and a further two groups online via Microsoft Teams (n= 9).” So different numbers of participants are mentioned.

Moreover, in the introduction section, I would suggest that you briefly mention some review of the literature on what characterizes a qualitative phenomenological study.

Line 89, “However, the benefits of drumming have not been explored with PLwP”, this is not true, among others you have the following example: DRUM-PD: The Use of a Drum Circle to Improve the Symptoms and Signs of Parkinson's Disease (PD) – movement disorder.

In the methods section please describe what type of qualitative phenomenological design did you choose (ie, descriptive, interpretative,…). Furthermore, were your participants screened for dementia before entering the program? Please clarify these in the document.

In the discussion section, line 289, “To the researchers' knowledge this is the first study which explored the impacts of cardio-drumming which has been adapted for PLwP” – please try not to be so expansive, since there are more studies, as previously stated.

In the limitations section, you mentioned “(JH & AW) were both study participants and research” – this implies that the anonymity of the participants is compromised, and this is not adequate in ethical terms.

Furthermore, the following text “Given the range of benefits reported in this study, future studies are warranted.” – I don’t see the need for such a statement, as it is not necessary nor adequate.

Lines 380-388 should be moved to the end of the conclusions section, the directions for future studies normally are described at the end of conclusions.

In the conclusions section, the following statement “Given the positive findings of this pilot study, future research is warranted”, again, I don’t think it is necessary or adequate, please consider removing this throughout the document.

Comments on the Quality of English Language

minor issues detected

Author Response

Many thanks for your comments. We made amendment according to your advice. We hope our manuscript is much improved. Thank you very much.

Round 2

Reviewer 2 Report (New Reviewer)

Comments and Suggestions for Authors

Dear authors, thank you for submitting the revised version.

The only thing that I would like to add is since much of the information regarding limitations has been deleted, would you please further elaborate on the limitations of your study?

Kind regards

Author Response

Dear Reviewer,

Many thanks for your help and support. We've amended the Limitation section, as you've suggested. Please see attached our responses & manuscript. We've elaborated our limitations, which is in yellow highlight. 

Much appreciated, 

Warmest regards,

Authors 

This manuscript is a resubmission of an earlier submission. The following is a list of the peer review reports and author responses from that submission.

Round 1

Reviewer 1 Report

Comments and Suggestions for Authors

Thank you for submitting your work. I have few suggestions that will make this work strong to get published. 1) Revise introduction for sentence structure and grammatical mistakes. Sentence 2 is starting with the number "10 million". You can write it as "Ten million". These simple mistakes are there in sentence structure; 2) For results section, can you please make a table of your findings of themes, in which participants are describing each feeling with common words used. Since there are only 7 men and 12 women, a table describing their feelings based on the themes will catch the audience on results faster; 3) Remove the unwanted gap between Theme 4 and 5. 

Author Response

Thank you so much for your support and comments. We've amended our manuscript accordingly. Please see the track changes in the doc and our responses below. Much appreciated.

1) Revise introduction for sentence structure and grammatical mistakes. Sentence 2 is starting with the number "10 million". You can write it as "Ten million". These simple mistakes are there in sentence structure;

Thank you for your comments. We’ve corrected it and edited the whole manuscript.

2) For results section, can you please make a table of your findings of themes, in which participants are describing each feeling with common words used. Since there are only 7 men and 12 women, a table describing their feelings based on the themes will catch the audience on results faster;

We have amended our Table 1 to include participants comments.

3) Remove the unwanted gap between Theme 4 and 5. 

Thank you. We’ve edited the whole manuscript with care.

Reviewer 2 Report

Comments and Suggestions for Authors

The Manuscript by Irons et al. titled " An exploration of people living with Parkinson’s experience of cardio-drumming, Parkinson’s Beats" aimed to study the effect of cardio-drumming on people living with Parkinson's (PLwP). The authors studied 19 PLwP in the regular Parkinson’s Beats sessions in-person or online and after 12 weeks, focus group discussions took place to understand the impacts of Parkinson’s Beats. The authors reported that the Participants involved in this study showed the range of benefits of cardio-drumming, including improved fitness and movement, positive mood, flow experience, and enhanced social well-being. In my opinion, the manuscript is interesting. I have some comments as follows:

- Authors mentioned that “In total, there were 7 men and 12 women, ranging between 55 and 80 years old, who took part in the Parkinson’s Beats programme”. The authors should provide the information (Age, sex, MoCA score, etc.) of the participants involved in this study.

- Authors should compare the effects of cardio-drumming/Parkinson’s Beats in men and women.

- Authors should improve the introduction with more updated information about the benefits of physical exercise in Parkinson’s.

- The number of participants involved in this study is not sufficient to conclude the results provided in the manuscript.

- The authors should discuss the cognitive status of the patients involved in this study. Were they from normal cognition or with mild cognitive impairment or dementia?

- Why did the authors not include healthy individuals in this study? Authors should compare this effect with healthy controls.

- Authors should provide a graphical representation for the overall conclusion of the manuscript. This will help readers to understand the overall conclusion of the manuscript.

Author Response

Many thanks for your support and comments. Please see our responses below and we've made amendments in the manuscript accordingly. We appreciate for your comments and time. 

 Authors mentioned that “In total, there were 7 men and 12 women, ranging between 55 and 80 years old, who took part in the Parkinson’s Beats programme”. The authors should provide the information (Age, sex, MoCA score, etc.) of the participants involved in this study.

- We have provided age, sex and their Parkinson’s-related details of our participants. We didn’t include the cognitive function screening using MoCA score, as this pilot study’s aim was to explore participants’ perspectives of the new cardio-drumming programme. Also, if a participant had declined cognitive function, they would haven’t been able to take part in both the sessions and the focus group discussion sessions. We haven’t observed any cognitive decline in our participants.

- Authors should compare the effects of cardio-drumming/Parkinson’s Beats in men and women.

- As stated above and in the manuscript, our aim of this pilot study was to learn from participants’ experience to gauge whether the cardio-drumming can be acceptable for people living with Parkinson’s. Thus, comparing the effects of Parkinson’s Beats between men and women is beyond the scope of this pilot, however, this may be a focus of future investigations.

- Authors should improve the introduction with more updated information about the benefits of physical exercise in Parkinson’s.

- Thank you for your comments. The references #11-#23 in the Introduction cover the benefits of exercise, which encompass a number of systematic review papers which date from 2016 to 2022, which offer currency to the literature used within the literature review.

 - The number of participants involved in this study is not sufficient to conclude the results provided in the manuscript.

- A total of 19 participants took part in the focus group discussions, which is a healthy number for a pilot, exploratory study. We’ve amended our conclusion to state that this is a pilot study. 

- The authors should discuss the cognitive status of the patients involved in this study. Were they from normal cognition or with mild cognitive impairment or dementia?

As stated above, we didn’t screen our participants in relation to the cognitive function. Also, if a participant had impaired cognitive function, they would haven’t been able to take part in both the cardio-drumming and the focus group discussion sessions. We haven’t observed any one with cognitive impairment in our study.

- Why did the authors not include healthy individuals in this study? Authors should compare this effect with healthy controls.

This was a pilot, exploratory study, as cardio-drumming is a new form of physical exercise for people living with Parkinson’s. As we indicated in the Discussion and Conclusion sections, further studies with control groups are needed to examine the effects of cardio-drumming.

- Authors should provide a graphical representation for the overall conclusion of the manuscript. This will help readers to understand the overall conclusion of the manuscript.

We understand that a graphical representation is optional. We’ll liaise with the editor regarding this.

Reviewer 3 Report

Comments and Suggestions for Authors

The Paper entitled “An exploration of people living with Parkinson’s experience of 2 cardio-drumming, Parkinson’s Beats by J. Yoon Irons1 and co-authors highlighted the beneficial effects of cardio-drumming (Parkinson’s Beats) on Parkinson’s disease-related symptoms in humans. In a small sample size, the authors have shown that Parkinson’s Beats significantly relieved the PD-associated symptoms. For the qualitative analysis, the authors have prepared a questionnaire, covering 6 basic themes and almost 16 sub-themes. The overall findings have suggested that Parkinson’s Beats is beneficial against PD-associated complications, covering movements and mode-related changes. This paper is a good addition to the studies related to the efforts working on Parkinson’s disease.

 My humble comments may be noted as under.

1.       The paper is nicely presented with quality writing.

2.       I was wondering if normal exercise is equally benefiting the patients, and how this difference was controlled. The authors mean that Parkinson’s Beats is better than normal exercise.

3.       The sample size is quite low, how can justify this?

4.       Most of the questions are not specific to PD. How these symptoms were just affixed with Parkinson’s disease.

5.       Overall, the paper has been nicely designed and efforts have been put into making it readable and understandable.

6.       Good luck

Comments on the Quality of English Language

good

Author Response

Thank you so much for your support and comments. Please see our responses below and track changes in the manuscript which show that we've amended our paper accordingly. 

  1. The paper is nicely presented with quality writing.

Thank you very much.

  1. I was wondering if normal exercise is equally benefiting the patients, and how this difference was controlled. The authors mean that Parkinson’s Beats is better than normal exercise.

This was a pilot, exploratory study, to assess whether the Cardio-drumming programme was acceptable to people living with Parkinson’s. Our aim was to learn from participants’ experience, so that a future study using robust methods could be developed. Our intention is to illustrate the benefits of this new form of exercise, cardio-drumming in the manuscript based on the focus group discussion data. As our findings suggest that it appears that there are some extra benefits from the cardio-drumming, such as experiencing the ‘flow’ and having music seems to be helpful too. We didn’t intend to compare the benefits of cardio-drumming with other types of exercises in this study.

  1. The sample size is quite low, how can justify this?

As our study was the first cardio-drumming study, we relied on volunteers. 19 participants are acceptable numbers for qualitative studies, as our aim was to explore participants’ perspectives of taking part in the cardio-drumming programme, Parkinson’s Beats.

  1. Most of the questions are not specific to PD. How these symptoms were just affixed with Parkinson’s disease.

We assume that “Most of the questions” refers to the focus group discussion questions. These questions are to explore participants’ experience in taking part in the very new cardio-drumming programme and the questions, which were open-ended with the intention to guide a group discussion.

  1. Overall, the paper has been nicely designed and efforts have been put into making it readable and understandable.

 Once again, many thanks for your comments.

Round 2

Reviewer 2 Report

Comments and Suggestions for Authors

The authors did not address all the comments. There are several points for which authors only gave some explanations but did not apply for improvement of this manuscript. For example, Healthy controls are required for comparison even if it is a pilot/exploratory study. However, authors ignored these comments.

Author Response

As above, we did address your concerns where possible. As we stated in our response #6 (yellow highlighted), our study was a pilot exploratory study without healthy controls. We repeatedly said that our study is a qualitative study, without a control group.

This study was not designed to explore the effectiveness of Parkinson's beats, rather the focus was to explore qualitatively what the perceptions of people with Parkinson's were in relation to this approach to exercise. As a qualitative study we were not aiming to look at differences, our focus was upon capturing the experiences and perceptions of the participants who had attended Parkinson's Beats. Therefore, it would not have been informative to interview people who have not participated in Parkinson's Beats, hence we did not include a control group.
